# Rapid label-free identification of seven bacterial species using microfluidics, single-cell time-lapse phase-contrast microscopy, and deep learning-based image and video classification

Erik Hallström[1]*, Vinodh Kandavalli[2], Carolina Wählby[1,3], Anders Hast[1]

1 Department of Information Technology, Uppsala University, Uppsala, Sweden, 2 Department of Cell and Molecular Biology, Uppsala University, Uppsala, Sweden, 3 SciLifeLab Science for Life Laboratory, Uppsala University, Uppsala, Sweden

* erik.hallstrom@it.uu.se

**Data availability statement:** The authors confirm that all data underlying the findings are

## Abstract

For effective treatment of bacterial infections, it is essential to identify the species causing the infection as early as possible. Current methods typically require hours of overnight culturing of a bacterial sample and a larger quantity of cells to function effectively. This study uses one-hour phase-contrast time-lapses of single-cell bacterial growth collected from microfluidic chip traps, also known as a "mother machine". These time-lapses are then used to train deep artificial neural networks (Convolutional Neural Networks and Vision Transformers) to identify the species. We have previously demonstrated this approach on four different species, which is now extended to seven common pathogens causing human infections: *Pseudomonas aeruginosa*, *Escherichia coli*, *Klebsiella pneumoniae*, *Acinetobacter baumannii*, *Enterococcus faecalis*, *Proteus mirabilis*, and *Staphylococcus aureus*. Furthermore, we expand upon our previous work by evaluating real-time performance as additional frames are captured during testing, and investigating the role of training set size, data quality, and data augmentation as well as the contribution of texture and morphology to performance. The experiments suggest that spatiotemporal features can be learned from video data of bacterial cell divisions, with both texture and morphology contributing to classifier decision. The method could be used simultaneously with phenotypic antibiotic susceptibility testing (AST) in the microfluidic chip. The best models attained an average precision of 93.5% and a recall of 94.7% (0.997 AUC) on a trap basis in a separate, unseen experiment with mixed species after around one hour. However, in a real-world scenario, one can assume many traps will contain the actual species causing the infection. Still, several challenges remain, such as isolating bacteria directly from blood and validating the method on diverse clinical isolates. This proof of principle study brings us closer to real-time diagnostics that could transform the initial treatment of acute infections.

fully available without restriction. A replication package is available at https://doi.org/10.5281/zenodo.13321089 containing all image data and software to reproduce the experiments, generate output metrics and build the graphs in the article.

**Funding:** Funding was awarded to C.W. from the Swedish Foundation for Strategic Research, https://strategiska.se/en, grant number: SSF ARC19-0016. The computations were enabled by the Berzelius supercomputing resource provided by the National Supercomputer Centre at Linköping University and supported by the Knut and Alice Wallenberg Foundation. Additionally, computations were enabled by the Alvis cluster provided by the National Academic Infrastructure for Supercomputing in Sweden (NAISS) at Chalmers University of Technology, partially funded by the Swedish Research Council through grant agreement no. 2022-06725. The funders had no role in study design, data collection and analysis, decision to publish, or preparation of the manuscript.

**Competing interests:** The authors have declared that no competing interests exist.

# 1 Introduction

## 1.1 Antibiotics and bacterial infections

Since its introduction in the mid-20th century, antibiotics have been pivotal for modern healthcare, saving countless lives [1]. It plays an integral role, not only in treating bacterial infections, but also in preventing them in patients with weakened immune systems, such as those undergoing chemotherapy for cancer treatment and transplant recipients [2]. Currently, antimicrobial resistance is on the rise [3–7], partly due to the overuse of broad-spectrum antibiotics [8,9]. Rapidly identifying the species causing the infection is necessary to mitigate this, as it helps select the best-optimized antibiotic therapy using AST outlined below, and reduces the likelihood of generating new resistance with antibiotic misuse. Also, for sepsis patients, prompt treatment with appropriate antibiotics is critical for survival, with every hour counting [10].

## 1.2 Antibiotic Susceptibility Testing

Antibiotic Susceptibility Testing (AST) refers to testing the efficacy of different antibiotics against bacterial infections. AST methods are broadly divided into genotypic and pheno-typic approaches. Genotypic AST detects specific genetic markers associated with antibiotic resistance, typically through quantitative Polymerase Chain Reaction (qPCR) assays targeting known resistance genes [11], sometimes simultaneously performing species identification and detecting resistance genes [12]. However, genotypic methods only detect targeted known resistance mechanisms and may miss novel or unexpected ones. Phenotypic AST assesses bacterial growth or survival directly in the presence of antibiotics—observing whether an organism is inhibited or killed by a drug. Phenotypic methods remain the gold standard as they capture all resistance mechanisms regardless of genotype.

Traditionally, phenotypic AST includes broth microdilution, typically performed in 96-well plates to determine the minimum inhibitory concentration (MIC), and the disk diffusion (Kirby–Bauer) method, where antibiotic disks are placed on inoculated agar plates. Although reliable, both methods require overnight incubation (typically 18–24 hours) and remain the most commonly used methods in clinical laboratories today [13].

Recent advancements aim to significantly accelerate phenotypic AST, since faster results can be lifesaving in critical infections like sepsis. By confining bacteria in microfluidic channels or nanoliter-scale droplets with antibiotics, even tiny amounts of growth or metabolic activity can be detected within 1-2 hours or less [14,15], contrary to waiting for macroscopic colony growth. One novel approach uses deep learning to rapidly evaluate microbial growth by analyzing angle-resolved scattered-light images of individual bacterial cells encapsulated in picoliter droplets within a microfluidic chip. [15]. A recent review [16] characterized over 90 next-generation AST platforms that achieve faster results than conventional methods, such as low-inoculum AST chip [17], nanomotion technology platform [18], flow cytometry [19], electrochemical sensing platforms [20], and laser scattering [21].

However, identifying the bacterial species before performing the AST is integral, as it determines which antibiotics are relevant to include in the susceptibility test, which is often species-specific. Furthermore, a very early indication of bacterial species can guide the choice of initial treatment.

## 1.3 Related species identification methods

Today, the primary bacterial identification method is Matrix-Assisted Laser Desorption/Ionization Time-of-Flight (MALDI-TOF) mass spectrometry [22]. Although very accurate, it requires overnight pre-cultivation (18-24 hours) on agar plates for large enough colonies to form. Several techniques have been explored to reduce the detection time. One possibility is using microscopy to analyze time-lapses of micro-colony growth, reducing the detection time to 6-12 hours [23,24]. Raman spectroscopy, illuminating bacteria samples with laser and analyzing the scattered wavelength spectrum, typically also requires a few hours for the microcolonies to form [25,26]. Also, genomic methods using multiplex qPCR can be used for bacterial identification [27] without pre-culturing, but they require prior knowledge of specific genetic targets and usually involve more complex sample preparation (turnaround time of 6 hours in the cited study) and carry a higher risk of contamination, especially when targeting a larger number of species. Another method for detection is using fluorescence microscopy with FISH (fluorescence in situ hybridization) [28]. While not requiring pre-cultivation, the method is destructive as the cell walls must be dissolved so the fluorescent stain can enter the bacteria, also, the number of species that can be detected in parallel is limited. In conclusion, although accurate, the genomic and spectroscopic detection methods remain complex, costly, and time-consuming.

In this paper, we investigate species identification based on spatio-temporal features from phase-contrast time-lapse microscopy of reproducing single cells. With few exceptions, clinically relevant bacterial species complete one cell division within an hour under optimal growth conditions. Using the *microfluidic mother machine* described below, the cells can be trapped and we can easily crop out images or video clips of a column of reproducing bacteria. This setup enables real-time, label-free identification of live, reproducing bacteria in under 70 minutes. Total turnaround time including isolation of cells would be 1-2 hours depending on the application, as demonstrated by previous publications using the microfluidic mother machine [14,28–30].

## 1.4 Microfluidics

Microfluidic devices for monitoring single-cell bacteria growth (example visualized in S9 Video) are fabricated with oblong traps with a width of around 1.5μm, laid out orthogonal to a larger center channel providing growth medium [31]. The traps accommodate only a single column of cells, with the mother cell located at the top of the trap, pushing the descendants down and eventually out into the center channel, hence the name "mother machine". A modification to this chip design introduced by Baltekin et al. [14] had a physical stop, protruding from the side of the trap wall, stopping the bacterial cells but allowing fluids to pass by. This allowed the microfluidic chip to be loaded faster using pressure techniques instead of diffusion methods used before (30s-10 min filling the device depending on cell culture density at 500mbar, 100mbar during running). Furthermore, growth medium or other substances, such as antibiotics, could be directly administered to the cells. This innovation led up to the founding of Sysmex-Astrego [32] and development of the PA-100 [30] system for phenotypic AST that is currently on the market.

## 1.5 Experiment setup

Data from this particular chip design introduced by Baltekin et al. [14] was used in this study. The overall experiment setup is shown in Fig 1. Note that fluorescence microscopy is only

## A. Microfluidic mother machine

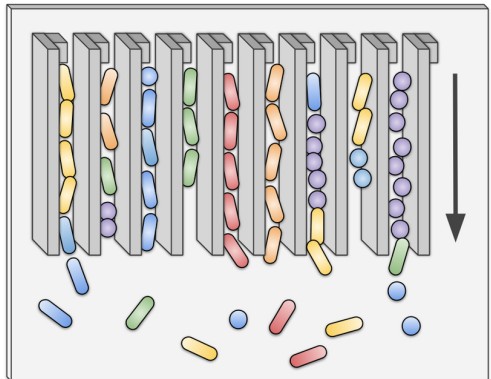

## B. Phase-contrast timelapse

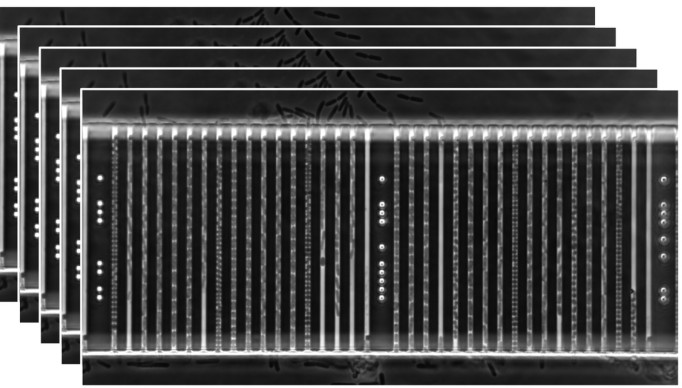

## C. Fluorescence

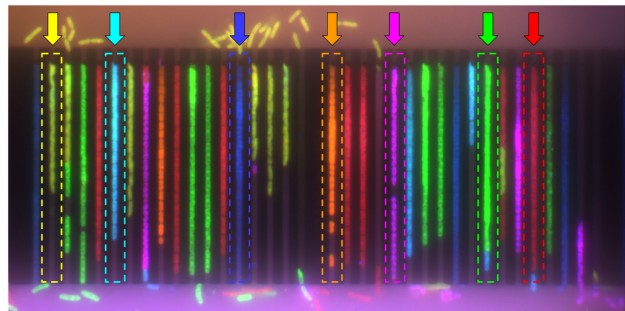

## D. Neural network

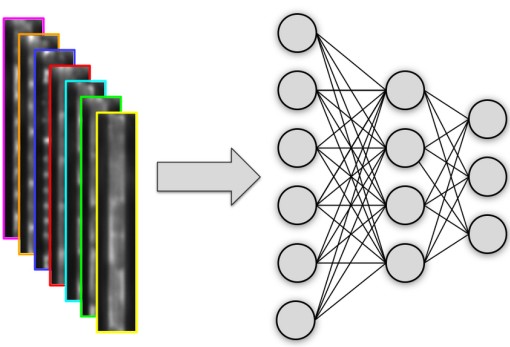

**Fig 1. Experimental setup.** A: Schematic of bacteria reproducing in a mother machine, the arrow on the right shows the growth direction. B: Phase-contrast time-lapse images of bacteria growing and reproducing in a mother machine. C: The fluorescence images captured after the final phase-contrast frame reveal the species in each trap (used as ground truth labels), with *E. coli* in green, *K. pneumoniae* in cyan, *E. faecalis* in magenta, *P. aeruginosa* in orange, *A. baumannii* in yellow, *P. mirabilis* in red and *S. aureus* in blue. Examples of cropping targets (traps containing a single species) for each of the seven species are outlined with dashed lines and pointed to by arrows in corresponding colors. D: Input to the neural network is phase-contrast only, a single frame or time-lapse video. Borders are color-coded according to the respective species.

used to obtain labels for the dataset; the classification is performed using phase-contrast time-lapses of living cells reproducing in the traps. A schematic of the bacteria reproducing in a mother machine is shown in Fig 1A, with the physical stop protruding out from the top-left side of the trap wall. Phase-contrast microscopy [33] is then used to collect a time-lapse containing single-cell growth inside the traps of the mother machine Fig 1B. After capturing the growth, cells are permeabilized, and fluorescently labeled probes binding to species-specific nucleic acid are administered to the cells for genotyping. Following the application of laser light, the cells emit light, revealing the species of bacteria in each trap. This procedure is referred to as fluorescence in situ hybridization (FISH) Fig 1C. Each trap is then cropped out, containing a phase-contrast time-lapse of bacterial growth. The fluorescence signal in the final frame provides the ground truth label. Using this data, we trained deep learning video or image classification models to classify the species growing inside each trap using only the phase-contrast data as input. The models were tested on one experiment with

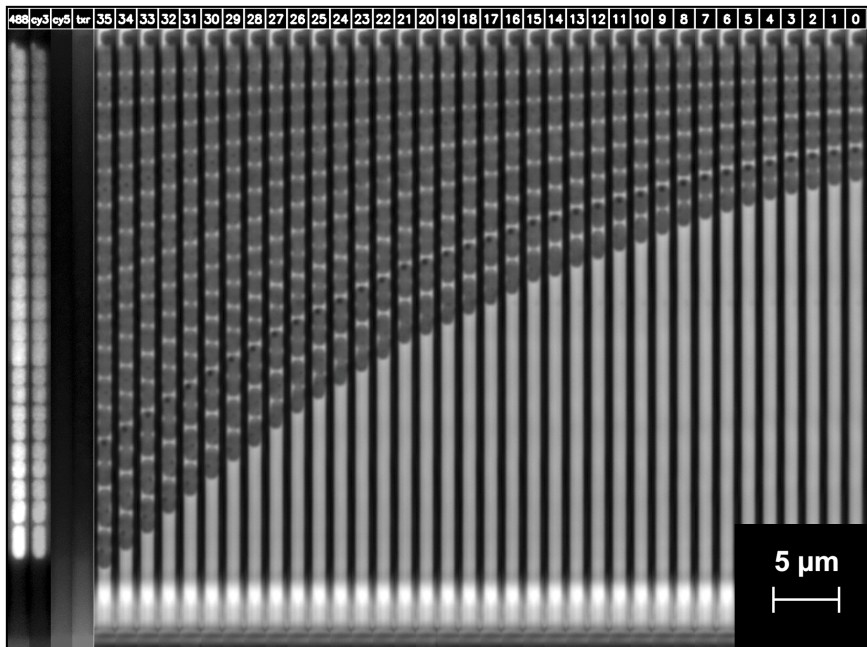

**Fig 2. One trap phase-contrast time-lapse and final fluorescence images.** Image of a cropped-out trap in the mother machine growing for 70 minutes and imaged every two minutes, resulting in 36 phase-contrast frames in the time-lapse (numbered 0-35), and the final fluorescence images for each channel are shown side-by-side. This sample is *A. baumannii* from Experiment 14 of the training set (see Table 1 in Materials and Methods).

mixed species not seen during training, illustrated in Fig 1D. We have previously demonstrated this technique for four species [34], using a separate fluorescent stain for each species. Recently Kandavalli et al. [28] have developed a method to identify seven species using fluorescence microscopy where the fluorescent probes attach to one or two species, combining the signal, where *E. coli* is visible in Cy3, *K. pneumoniae* in Cy3 and Cy5, *E. faecalis* in Alexa488, *P. aeruginosa* in Alexa488 and Txr, *A. baumannii* in Alexa488 and Cy3, *P. mirabilis* in Alexa488 and *S. aureus* in Cy5 and Txr. Thus the dataset contains two cocci species with a circular shape and five rod-shaped species. This study uses the dataset available from this publication for testing and evaluation. A phase-contrast microscopy time-lapse followed by the four fluorescence frames of one cropped-out trap with *A. baumannii* is shown Fig 2. The full procedure at a single position in the microfluidic chip is shown in S9 Video.

## 1.6 Dataset and deep learning models

Since the early 2010s, deep artificial neural networks (referred to as "deep learning") [35] have emerged as the most prominent technique for analyzing data across many modalities such as images [36,37], text [38,39], video [40] and audio [41]. In this study, both image and video classification methods were trained and evaluated from the two main families, Vision Transformer (ViT) [37] and Convolutional Neural Network (CNN) [42,43]. The models were trained on a phase-contrast dataset of 18101 traps obtained from 16 experiments and tested on a separate test experiment containing 1399 traps that were not seen during training. Each trap contains single-species growth captured with phase-contrast microscopy, around 1-hour

time-lapses, 2 minutes per frame. Traps with mixed species, as seen in Fig 1C were discarded as they are unlikely to occur in a real-world setting. The whole dataset consists of 621,931 images. The best model attained a precision of 93.5% and a recall of 94.7% (0.997 AUC) on a trap basis within 60 minutes. Furthermore, the performance was only slightly degraded when downsampling, allowing it to be implemented in a potential low-resource clinical setting.

## 2 Results

### 2.1 Models and evaluation metrics

In this study, two similarly sized CNN-based video classification networks were tested: R(2+1)D with 31.5M parameters and R3D Video ResNet with 33.4M parameters from [40]. In addition, three CNN-based image classification networks: ResNet 18 [36] with 11.7M parameters, RegNetY 016 [44] with 11.2M parameters, EfficientNet B3 [45] with 12.23M parameters, and one ViT-based FastViT SA12 with 11.6M parameters [46]. The entire trap had a width of 52 pixels and a height of 1500 pixels, but the spatial input to the models was set to 52x114 pixels. The growth extent in each trap could be obtained using a semantic segmentation model. Using this vertical growth, a number of crops, either still images (S1 Fig) or video clips (S2 Fig), could be extracted in a tiling window fashion. The final output for a trap was obtained by summing the output logits (unnormalized network output) from each crop (image or video clip).

The evaluation metrics used were precision, recall, one-vs-rest and micro-averaged Area Under the Curve (AUC) (see Metrics in Materials and Methods section). One unseen experiment was used for testing data, from which 447 *A. baumannii*, 287 *K pneumoniae*, 175 *S. aureus*, 154 *P. mirabilis*, 116 *E. coli*, 139 *E. faecalis*, and 81 *P. aeruginosa* traps were extracted. The test set had 35 phase-contrast frames captured during 70 minutes.

The networks were retrained for each setting with a set random seed to ensure reproducibility: five times for the scaling experiments (except for the data scaling experiment) and 30 times for the categorical experiments. Each point in the following visualization represents evaluation on a separately trained neural network. The lines are added to track the mean values in the scaling experiments and are not meant for interpolation or statistical inference. Vertical lines in the categorical plot show standard deviation.

### 2.2 How many frames do we need, and can we learn single-cell spatiotemporal features?

Model performances were tested by gradually increasing the frames available in the time-lapse, each frame corresponding to 2 minutes, shown in Fig 3. Using video classification is better than aggregating still images after a few minutes. In this case, ResNet 18 outperforms the modern FastViT. No substantial improvements were observed using the other more modern CNN designs. The video classification network reaches 0.99 AUC within 40 minutes. Notably, spatiotemporal features are important for differentiating the rod species *E. coli*, *K. pneumoniae*, and *P. mirabilis*.

A confusion matrix of the Video ResNet R(2+1)D model tested using the final full 35-frame time-lapse is shown in Fig 4. A few misclassifications of this model are shown in S3–S8 Figs.

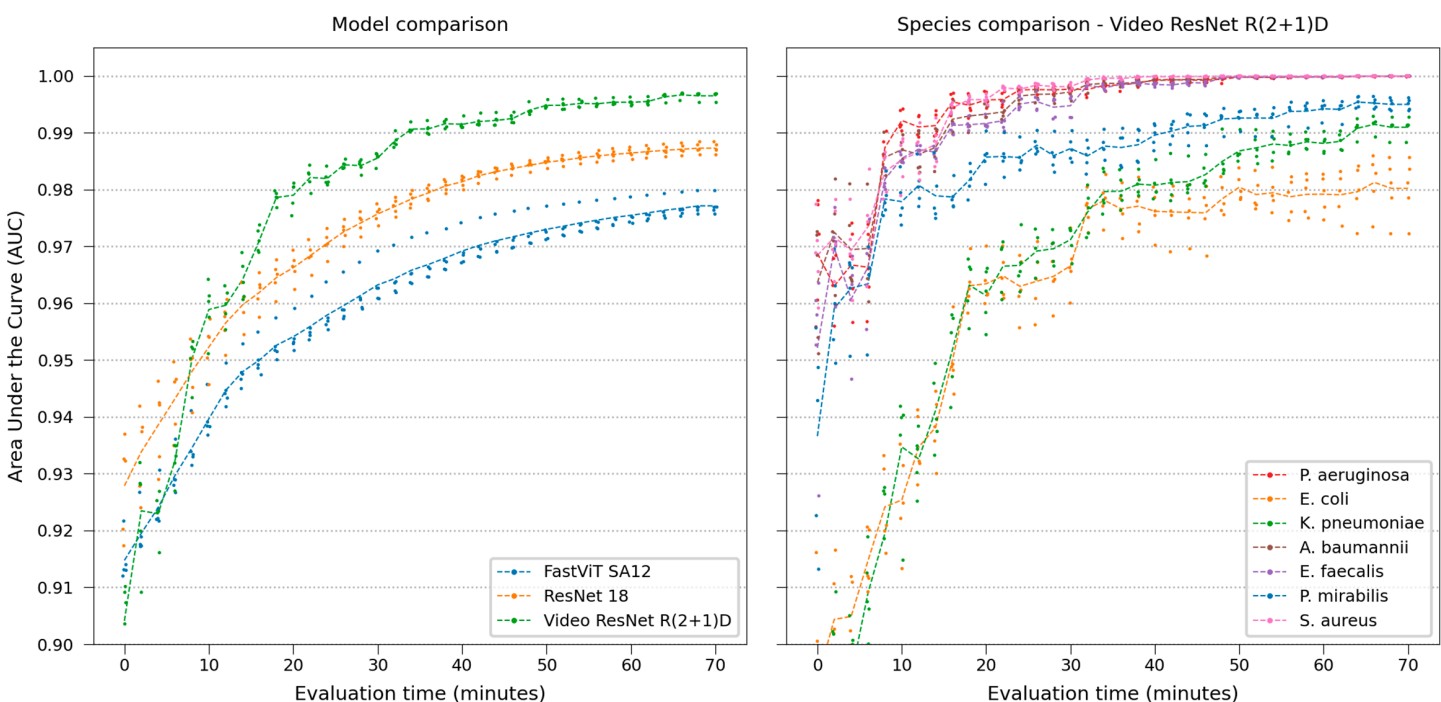

**Fig 3. Real-time performance.** Comparing model and Video ResNet R(2+1)D species classification performance over time, gradually using more frames in the time-lapse.

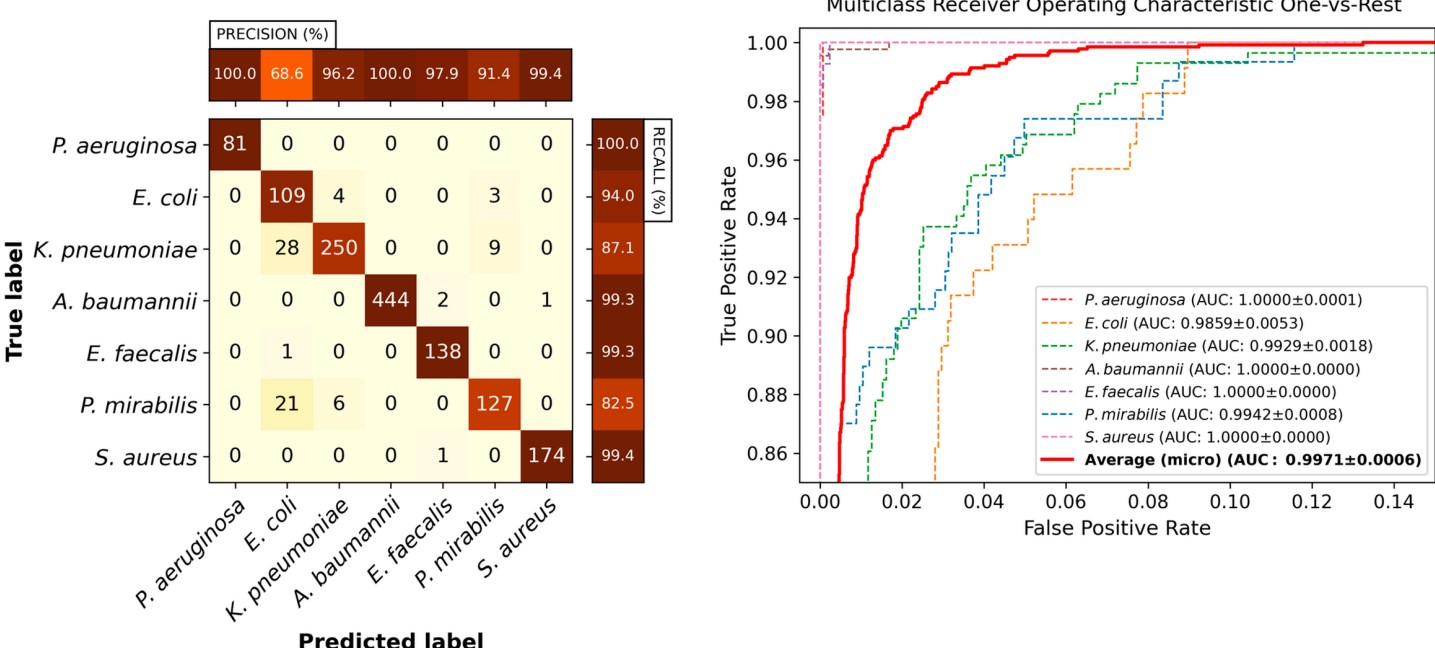

**Fig 4. Video classification performance.** Classification performance of Video ResNet R(2+1)D utilizing the full crop size 52x114 pixels, using all 35 frames (full time-lapse). Left: Confusion matrix for the classifier with the median AUC across the five retrainings. Right: Receiver operating characteristic curves for the same model. Note that the plot has been zoomed in for visibility. The legend displays the mean and standard deviation of the AUC across the five retrainings.

## 2.3 Does it work on spatially downsampled frames?

The networks were retrained on spatially downsampled images to simulate use in lower-resource settings, using Lánczos [47] downsampling. This serves two purposes: first, smaller images reduce computational and memory requirements at inference time, making deployment on a clinical end device more feasible. Second, downsampling can approximate the effect of using a lower-resolution microscope. While Lánczos is designed to preserve high-frequency details and does not replicate the optical blur introduced by lower numerical apertures—which is primarily determined by the objective lens—nor the true sampling characteristics set by the sensor pixel size, it can still serve as a useful proxy or proof of concept for evaluating model robustness under degraded imaging conditions. In total, 28 downsampling steps were used from 52x114 pixels to 3x7 pixels, visualized in S1–S7 Videos. The evaluation was performed on the whole time-lapse (35 frames representing 70 minutes). Video ResNet R(2+1)D again attained the best performance, AUC only dropping to 0.96 when using 35-frame time-lapses at 3x7 pixels (very low resolution) shown in Fig 5.

A confusion matrix of the Video ResNet R(2+1)D model trained using 35-frame time-lapses operating at 5x11 pixels is shown in Fig 6, indicating an increasing number of rod-shaped species confused. A few misclassifications of this model are shown in S9–S14 Figs.

Furthermore, time evaluations were performed to assess the performance degradation of Video ResNet R(2+1)D, training and testing using gradually more spatially downsampled time-lapses, shown in Fig 7.

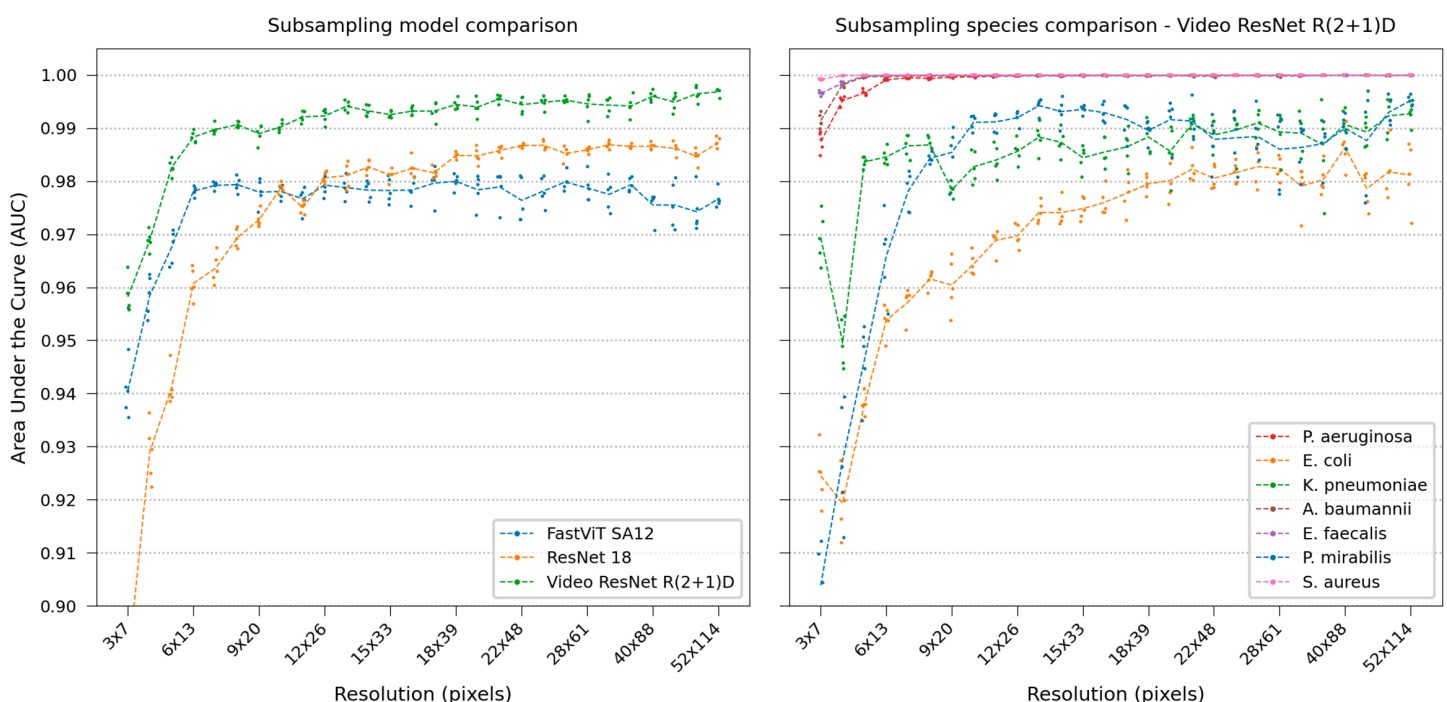

**Fig 5. Model comparison on downsampled images.** Retraining the networks with the image input spatially downsampled during test and training, testing on the whole time-lapse. Left: Model comparison. Using spatiotemporal features (video classifier) is especially important at very low resolution. Right: Species comparison using Video ResNet R(2+1)D at 5x11 pixels. Four species attain classification performance of 0.99 AUC even at very low resolution.

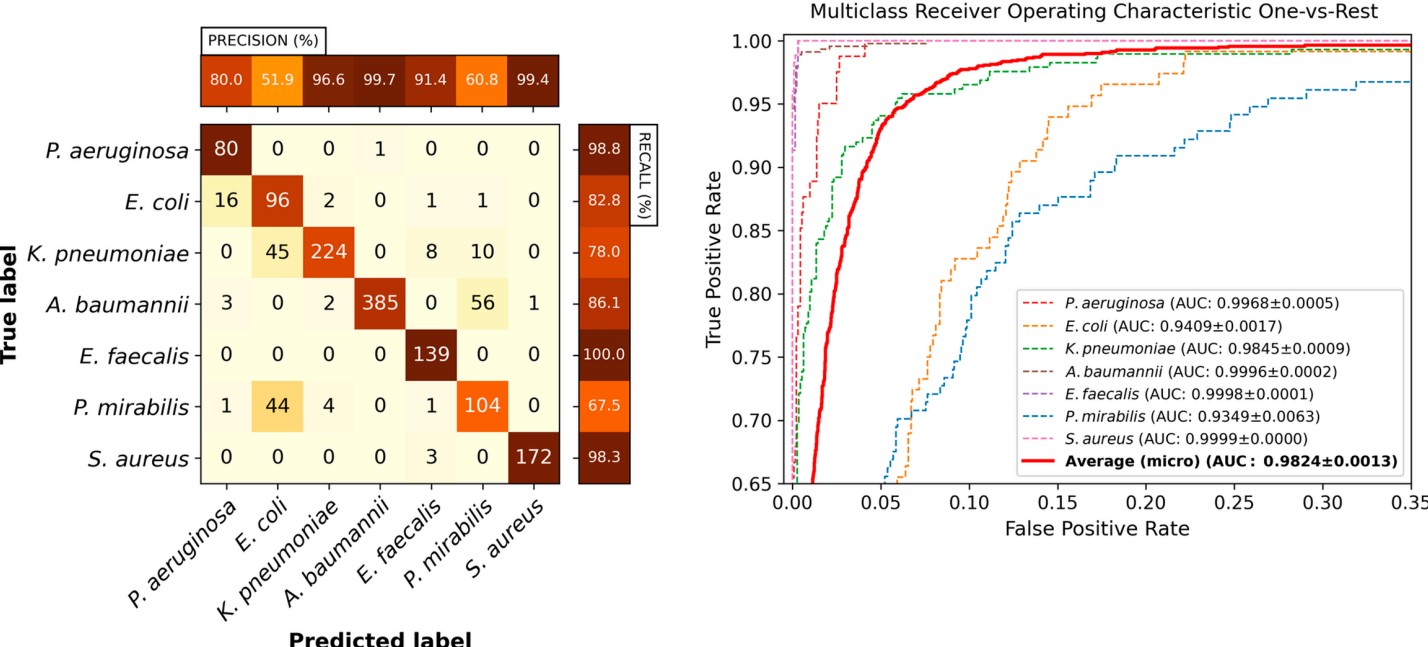

**Fig 6. Video classification performance at 5x11 pixels.** Classification performance of Video ResNet R(2+1)D utilizing spatially downsampled images 5x11 pixels, using all 35 frames (full time-lapse). Left: Confusion matrix for the classifier with the median AUC across the five retrainings. Right: Receiver operating characteristic curves for the same model. Note that the plot has been zoomed in for visibility. The legend displays the mean and standard deviation of the AUC across the five retrainings.

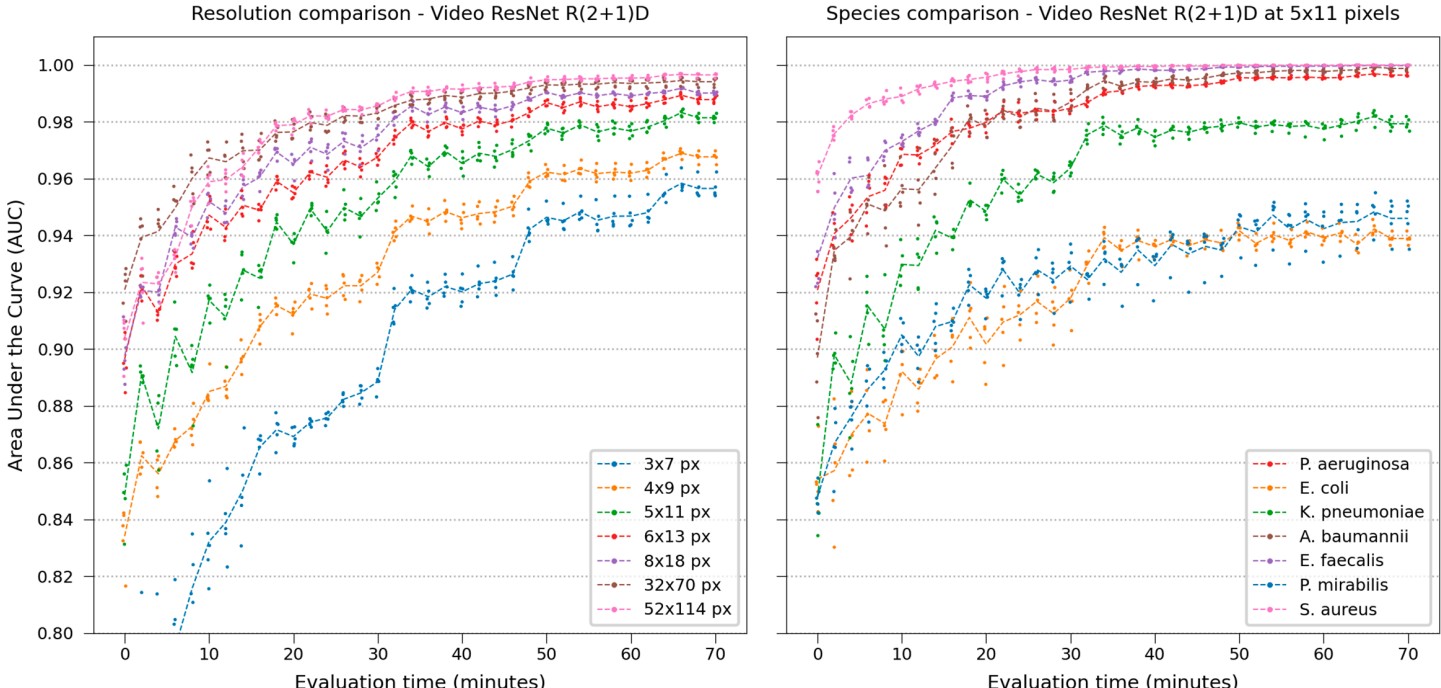

**Fig 7. Classification performance at various downsampling steps.** Left: Degradation of performance during time evaluations using Video ResNet R(2+1)D trained with spatially downsampled data. Distinct performance jumps occur at specific time points for the highly downsampled data, highlighting the significance of spatiotemporal features at low resolutions. Right: Time evaluation of Video ResNet R(2+1)D operating at spatially downsampled crop size of 5x11, visualizing species-specific AUC. Note that the y-axis of this plot starts at 0.8AUC.

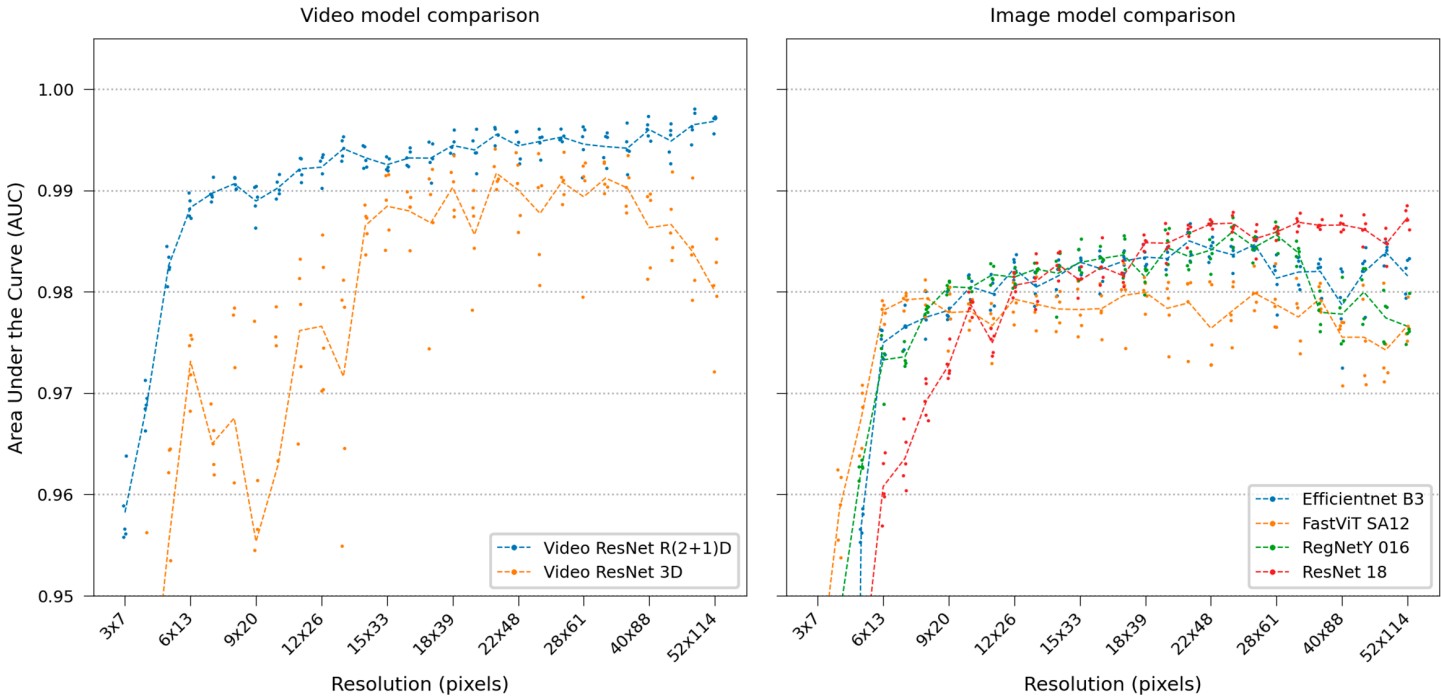

**Fig 8. Video classification performance at various downsampling steps.** Model comparison between video and image classification at gradually lower resolutions. Video classification performs better, especially at lower resolutions.

A comparison of all video and image classification models is shown in Fig 8, notably Video ResNet R(2+1)D using decomposed convolutions in spatial (2D) and temporal (1D) domains performs better and is more stable than Video ResNet 3D utilizing 3D convolutions (see Deep learning model training in Materials and Methods for details). There is no apparent improvement when using more modern CNN designs such as EfficientNet, RegNetY, or the modern ViT model, FastViT.

## 2.4 What augmentations are important?

Several augmentation methods from the Albumentation library [48] were used during the training process. Additionally, two custom developed methods were employed: Random video frame erasing (setting all pixels to zero in randomly selected frames in the time-lapse) and Segment (using the segmentation model to erase the background). A range of experiments were performed using Video ResNet R(2+1)D, removing one of these augmentations during training. In one experiment, the models were trained from scratch with randomly initialized weights instead of using pretrained weights used by default. Finally, one experiment was conducted removing all training augmentations.

Visualized in Fig 9, RandomBrightnessContrast had the largest impact on performance, followed by Label smoothing [49] (see Augmentations in Materials and Methods for explanation), suggesting that the models easily overfit to experiment settings (such as illumination and contrast). Removing all augmentations was detrimental to the performance.

**Fig 9. Removing augmentation.** The performance was tested by removing one augmentation and retraining the Video ResNet R(2+1)D. The baseline with all augmentation present is shown using a dashed red line. The plots are split up for readability due to the removal of RandomBrightnessContrast having a larger impact. Also, using label smoothing was beneficial, preventing the networks from overfitting. The removal of all augmentations experiment used randomly initialized weights and no label smoothing.

## 2.5 Can the classifiers learn texture, morphology, and division pattern?

The full-resolution Video ResNet R(2+1)D and ResNet 18 models were retrained and evaluated using only pre-generated segmentation masks (see Segmentation model in Materials and Methods). Hence, texture information was removed. Then, this was repeated, but instead replacing each pixel in the images with the row-wise average, meaning that the models could only see division patterns and cell length (refer to Image Modifications in Materials and Methods for more details).

The results are visualized in Fig 10. The video model, which can observe growth and division speed through row-wise mean modifications, has a much smaller performance drop than the still-frame image classification model, which can only infer cell length from the static images.

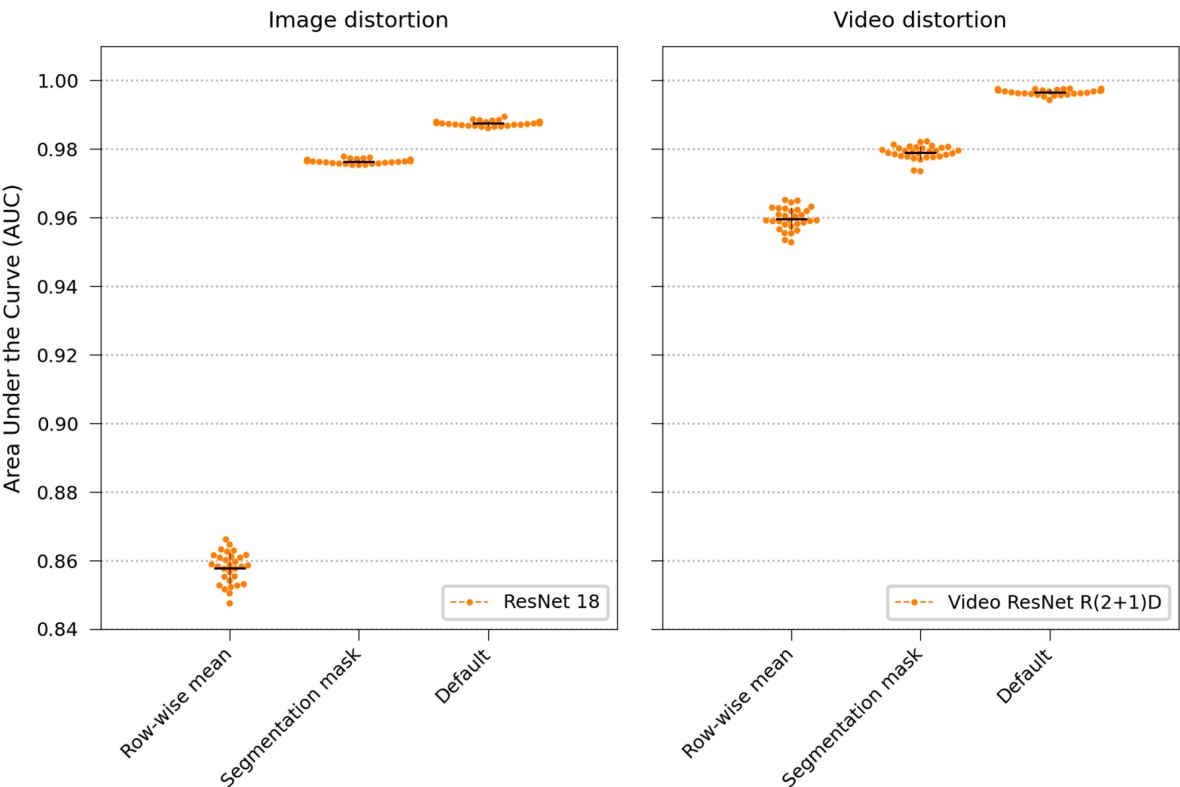

**Fig 10. Distortions.** Assessing the significance of texture, morphology, cell length, and growth and division speed by comparing static images classified using ResNet 18 with video clips classified using Video ResNet R(2+1)D.

## 2.6 How does the quality of the training data affect performance?

We created a custom data-quality metric that assigned a score to each trap in the training set, favouring traps with steady growth without abrupt movements. Networks were trained using three different training set sampling strategies outlined in "Data quality score" in Materials and methods, results shown in Fig 11. Using the quality metric was beneficial for both the video and image models, hence all experiments in this study was conducted limiting the maximum number of traps per species to 2600, selected based on the quality score. The quality score was not used for the test set.

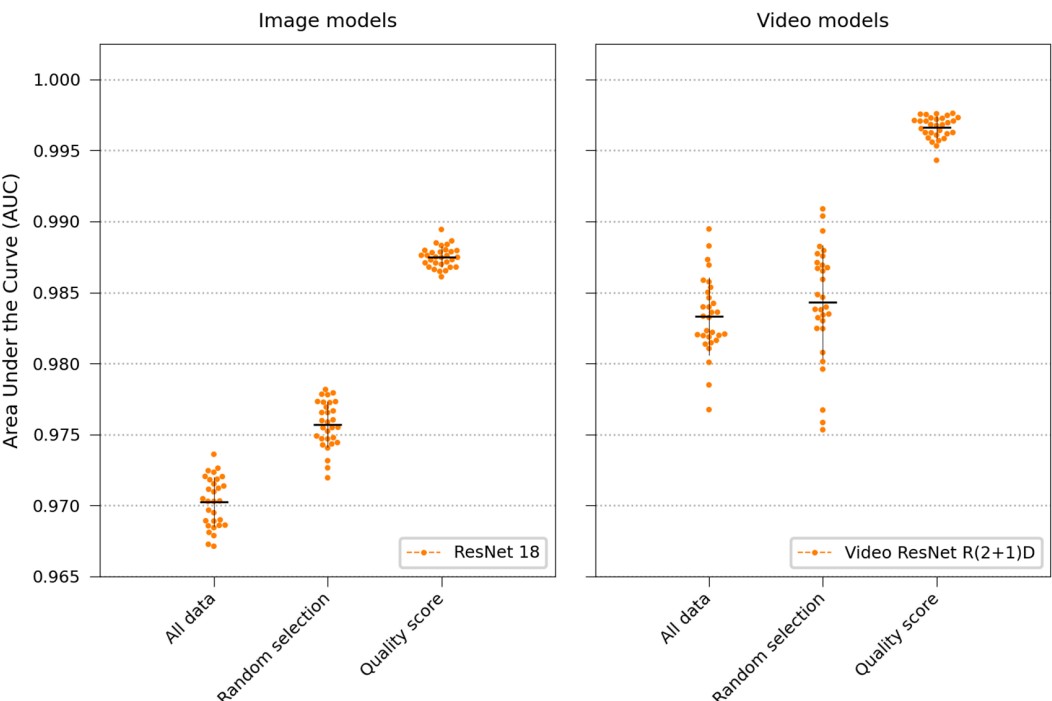

**Fig 11. Results using data quality scoring.** The ResNet 18 and Video ResNet R(2+1)D networks were retrained 30 times using three different approaches: (1) using all data in the training set but utilizing a random weighted sampler, (2) limiting the number of traps per species to 2600 using random selection, (2) limiting the number of traps per species to 2600 but using data quality score to select the traps.

## 2.7 How much data do we need?

Data scaling experiments were performed using Video ResNet R(2+1)D and ResNet 18 and training on downsampled images at 12x26 pixels and the original resolution 52x114 pixels. The network was retrained multiple times with a progressively larger training set by increasing the number of included traps per species, keeping all other settings fixed. It should be noted that the traps were sorted based on the data quality score from Algorithm 1 (see Label Adjustments in Materials and Methods), thus high-quality samples were included first when scaling. The networks were also trained without data augmentation. As shown in Fig 12, augmentation was essential: without it, performance plateaued and overfitting emerged as dataset size increased (see the Discussion for possible reasons).

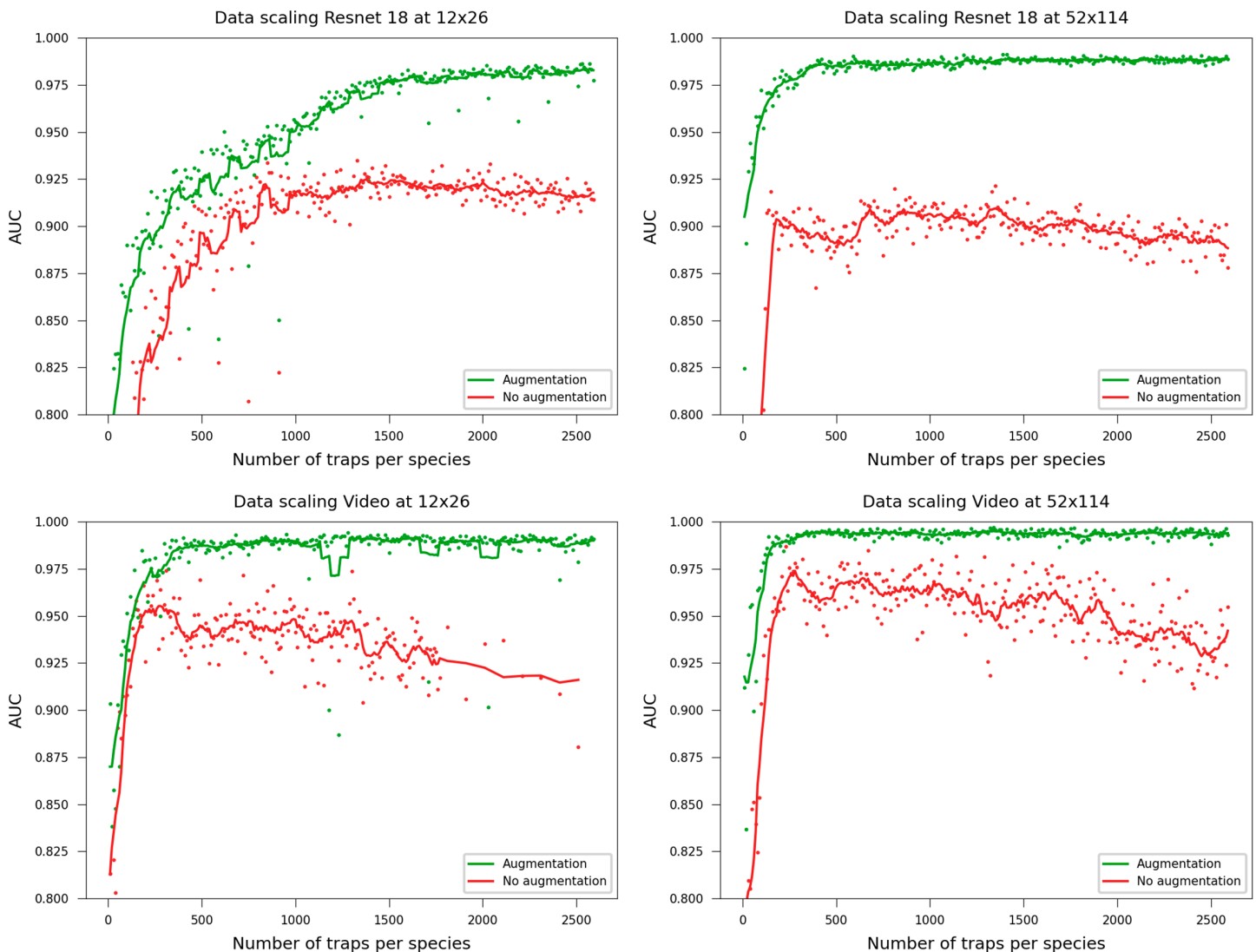

**Fig 12. Data scaling.** Performance when including more traps per species in the dataset. The lines are added moving averages with a rolling window size of 5 to track the general trend. Augmentation was essential across all dataset sizes for the model's ability to generalize.

## 3 Discussion

### 3.1 Developments compared to our previous work

Our previous study [34] used seven experiments, each with four mixed species loaded into each chip section, testing on a random 15% train-test split (splitting on a trap basis). The setup presented in this paper was more challenging, the training set had only 1-2 species per chip and chip section, testing was performed using a separate experiment with mixed species in both sections, as seen in Table 1. Also, *K. pneumoniae* and *E. coli* were more distinguishable in our previous study (*K. pneumoniae* being slightly thicker and bulkier). This study introduced two new rod species, *P. mirabilis*, and *A. baumannii*, which were visually

almost identical to *E. coli* and *K. pneumoniae*. Thus, the networks must learn subtle features not clearly visible to the human eye.

Furthermore, this study extends our previous work by evaluating real-time classifier performance as a function of the number of observed time-lapse frames, which would correspond to minutes in a potential clinical setting (Fig 3), and by investigating the effects of training set size (Fig 12), data quality (Fig 11), augmentations (Fig 9), and testing how texture and morphology contribute to classifier decision (Fig 10). The experiments suggest that both morphology, textures, and spatiotemporal features contribute to the final performance of the models and that learning spatiotemporal features is especially important at lower resolutions. Using augmentation and label smoothing during training greatly improved the performance, indicating the models easily overfitted to irrelevant patterns in the training set (experiment settings), possibly due to the fact that models were trained on data from experiments with only 1-2 species per chip.

Our hypothesis for the data scaling experiments (Fig 12) is that, provided the models are not allowed to overfit to patterns in the training set, the intra-species variability is low, likely because the samples are lab isolates. Only a few hundred samples per species are required before performance plateaus at 0.98 AUC. This is further supported by the observation that increasing the dataset size without augmentation led to worse results, similar to overfitting effect when increasing the number of training epochs of a fixed size dataset. Note that the number of pixels the models and thus the convolutional filters can see during training is $num\_epochs \times width \times height \times num\_frames \times num\_traps\_per\_species \times 7$. This means that ResNet 18 training on downsampled images saw only $1.7 \cdot 10^9$ pixels, compared to Video ResNet at full resolution, which saw $161 \cdot 10^9$ pixels during training. There is possibly an intricate relationship between augmentation, number of training samples and resolution.

Interestingly, modern neural network architectures did not outperform ResNets in this study. Similarly to our previous study, a significant proportion of the performance is retained at low resolution, especially when using video classification networks.

## 3.2 Limitations and future work

We would like to emphasise that this study is a proof of concept. The model was trained and evaluated on laboratory strains of bacteria. Several classes of major pathogenic species, such as Gram-negative cocci and Gram-positive rod-shaped cells, were not included. Future work would investigate the methodology using a broader range of bacterial species and clinical isolates from many different patients, as these can exhibit varied phenotypes. Also, the culture medium may affect the optical phenotype.

Our proposed species identification method requires the bacteria to be separated from other cell types. For urine samples, screening could be applied directly, as demonstrated by the PA-100 [30] system developed by Sysmex-Astrego [32], where the phenotypic susceptibility testing could be extended to perform species identification. Since phenotypic AST relies on sensor-based detection of bacterial growth or presence of living cells, one could envision some of the novel AST systems mentioned in the Introduction section performing both tasks, such as using 2D scattering images in [15] for species identification [50].

For blood samples, however, our method requires prior separation of bacteria from larger blood cells to prevent clogging of the microfluidics. A novel approach for such separation is outlined by Henar et al. [29]

Although this study utilizes a specific microfluidic chip, the "mother machine," one can envision classifying bacterial species based on spatiotemporal features from single-cell growth in other settings captured through alternative image modalities. This enables future work to extend beyond the particular setup used in this paper.

## 3.3 Clinical impact

The methodology can be applied prior to or concurrently with phenotypic AST testing and give an early warning before results of other approaches to species identification, such as MALDI-TOF, arrive from the medical laboratory. Additionally, the growth can be monitored for an extended time longer than one hour, updating the estimate as time progresses.

The study presented in this paper further demonstrates the potential of training deep learning models to identify bacterial species in a much shorter time than routine methods today. Instead of days to obtain positive cultures, typing, and phenotypic drug sensitivity testing results, these new methods decrease the waiting time to hours or even minutes. Such developments could lead to a paradigm shift in how to treat acute bacterial infections, moving from the current "try-and-test" approach to more knowledge-based decisions. This could improve patient outcomes, minimize antibiotic misuse, and ultimately save lives.

## 4 Materials and methods

The following sections outline the main components of the software and data. However, for full implementation details, refer to the released replication package [51], which contains documentation, software and all image data to reproduce the experiments, as well as pre-trained models. Python version 3.10.12 was used for all software, apart from the libraries stated below the software utilized NumPy [52], Pandas [53], and Matplotlib [54].

## 4.1 Microfluidics, imaging, bacterial strains and biological details

The dataset was obtained from Kandavalli. et al. [28], for detailed information on the biological experiment, please refer to their publication. The details relevant for the research presented here are restated below for convenience.

**4.1.1 Bacterial strains.** Seven common pathogens were selected that cause human infections, including Gram-positive cocci: *S. aureus* (ATCC 29213) and *E. faecalis* (ATCC 51299), and Gram-negative rods: *E. coli* K12 MG1655 (DA4201), *K. pneumoniae* (ATCC 13883), *P. mirabilis* (ATCC 29906), and *A. baumannii* (DA68153), *P. aeruginosa* (DA6215).

**4.1.2 Media and culture conditions.** All strains were grown overnight at 37C in Mueller-Hinton (MH) medium with shaking (200 rpm). The overnight culture was diluted 1:1000 in fresh MH medium supplemented with Pluronic F-108 (0.085% wt/vol) and grown for 2h under the same conditions. Equal amounts of each strain were then mixed and loaded onto the microfluidic chip.

**4.1.3 Microfluidics and experimental setup.** The microfluidic chip consisted of a micro-molded polydimethylsiloxane (PDMS) layer (Sylgard 184) covalently bonded to a 1.5 glass coverslip (Menzel-Gläser), as described in [14]. Specific ports were used for cell loading, media exchange, and back-channel pressure control. Pressure was maintained at 200 mbar using an OB1-Mk3 pressure regulator (Elveflow).

## 4.2 Microscopy imaging

A Nikon Ti2-E inverted microscope with a Plan Apo Lambda 100× oil immersion objective and a DMK 38UX304 camera was used for phase contrast and fluorescence imaging, same optical setup as in [14].

## 4.3 Dataset

The data consisted of 17 different experiments, each loading a microfluidic chip with a bacterial sample of 1-7 species and monitoring bacterial growth for 58-74 minutes, where an image frame was captured every 2 minutes, thus 30-36 phase-contrast frames in each time-lapse. The phase-contrast microscopy images, shown in Fig 1B, and final fluorescence images, shown in Fig 1C, were captured at each position on the chip where a position consisted of 42 traps and had an image size of around 2000x4000 pixels (one position also shown in S9 Video). Each trap was cropped out using a cropping program outlined below, resulting in crops with an image size of 52x1500 pixels, 30-36 phase-contrast frames, and four fluorescence frames per crop. Empty traps and traps containing multiple species were discarded using a labeling tool described below. Each microfluidic chip contained two sections, "Up" and "Down", where a section could be loaded with a specific species or a mix of species. The crosstable of the dataset used for training the neural networks is shown in Table 1, and the corresponding number of traps of each species extracted from each experiment. A number of sample images from the dataset are shown in Fig 13.

**Table 1. Data summary.**

|  | Frames | *A. baumannii* Up | Down | *E. coli* Up | Down | *E. faecalis* Up | Down | *K. pneumoniae* Up | Down | *P. mirabilis* Up | Down | *P. aeruginosa* Up | Down | *S. aureus* Up | Down | All species Total |
|---|---|---|---|---|---|---|---|---|---|---|---|---|---|---|---|---|
| Experiment 1 | 33 | 0 | 0 | 0 | 0 | 0 | 0 | 415 | 466 | 0 | 0 | 0 | 0 | 0 | 0 | 881 |
| Experiment 2 | 32 | 0 | 0 | 0 | 0 | 0 | 0 | 0 | 0 | 0 | 0 | 955 | 1169 | 0 | 0 | 2124 |
| Experiment 3 | 32 | 0 | 0 | 0 | 0 | 136 | 1211 | 0 | 0 | 0 | 0 | 0 | 0 | 0 | 0 | 1347 |
| Experiment 4 | 31 | 0 | 0 | 733 | 830 | 0 | 0 | 0 | 0 | 0 | 0 | 0 | 0 | 0 | 0 | 1563 |
| Experiment 5 | 30 | 0 | 0 | 0 | 21 | 0 | 0 | 40 | 0 | 0 | 0 | 0 | 0 | 0 | 0 | 61 |
| Experiment 6 | 30 | 0 | 0 | 0 | 0 | 390 | 0 | 0 | 0 | 0 | 0 | 0 | 47 | 0 | 0 | 437 |
| Experiment 7 | 32 | 0 | 183 | 0 | 0 | 0 | 0 | 0 | 0 | 528 | 0 | 0 | 0 | 0 | 0 | 711 |
| Experiment 8 | 30 | 0 | 0 | 0 | 0 | 0 | 0 | 0 | 0 | 0 | 0 | 0 | 0 | 656 | 664 | 1320 |
| Experiment 9 | 30 | 0 | 0 | 134 | 128 | 66 | 156 | 0 | 0 | 0 | 0 | 0 | 0 | 0 | 0 | 484 |
| Experiment 10 | 30 | 0 | 0 | 0 | 0 | 0 | 0 | 813 | 866 | 0 | 0 | 0 | 0 | 520 | 670 | 2869 |
| Experiment 11 | 35 | 442 | 604 | 0 | 0 | 0 | 0 | 0 | 0 | 0 | 0 | 1 | 66 | 0 | 0 | 1113 |
| Experiment 12 | 30 | 0 | 0 | 0 | 64 | 0 | 0 | 0 | 0 | 572 | 651 | 0 | 0 | 0 | 0 | 1287 |
| Experiment 13 | 29 | 0 | 0 | 49 | 45 | 79 | 211 | 0 | 0 | 0 | 0 | 0 | 0 | 0 | 0 | 384 |
| Experiment 14 | 36 | 674 | 697 | 0 | 0 | 0 | 0 | 0 | 0 | 0 | 0 | 120 | 242 | 0 | 0 | 1733 |
| Experiment 15 | 31 | 0 | 0 | 179 | 174 | 94 | 257 | 0 | 0 | 0 | 0 | 0 | 0 | 0 | 0 | 704 |
| Experiment 16 | 31 | 0 | 0 | 0 | 243 | 0 | 0 | 0 | 0 | 399 | 441 | 0 | 0 | 0 | 0 | 1083 |
| **Experiment 17** | **35** | **210** | **237** | **53** | **63** | **73** | **66** | **118** | **169** | **60** | **94** | **37** | **44** | **69** | **106** | **1399** |
| All experiments |  | 1326 | 1721 | 1148 | 1568 | 838 | 1901 | 1386 | 1501 | 1559 | 1186 | 1113 | 1568 | 1245 | 1440 | 19500 |

Time-lapses extracted from all seventeen experiments; each entry corresponds to a microfluidic trap containing only one species of bacteria. The last experiment highlighted and marked in bold was used for testing.

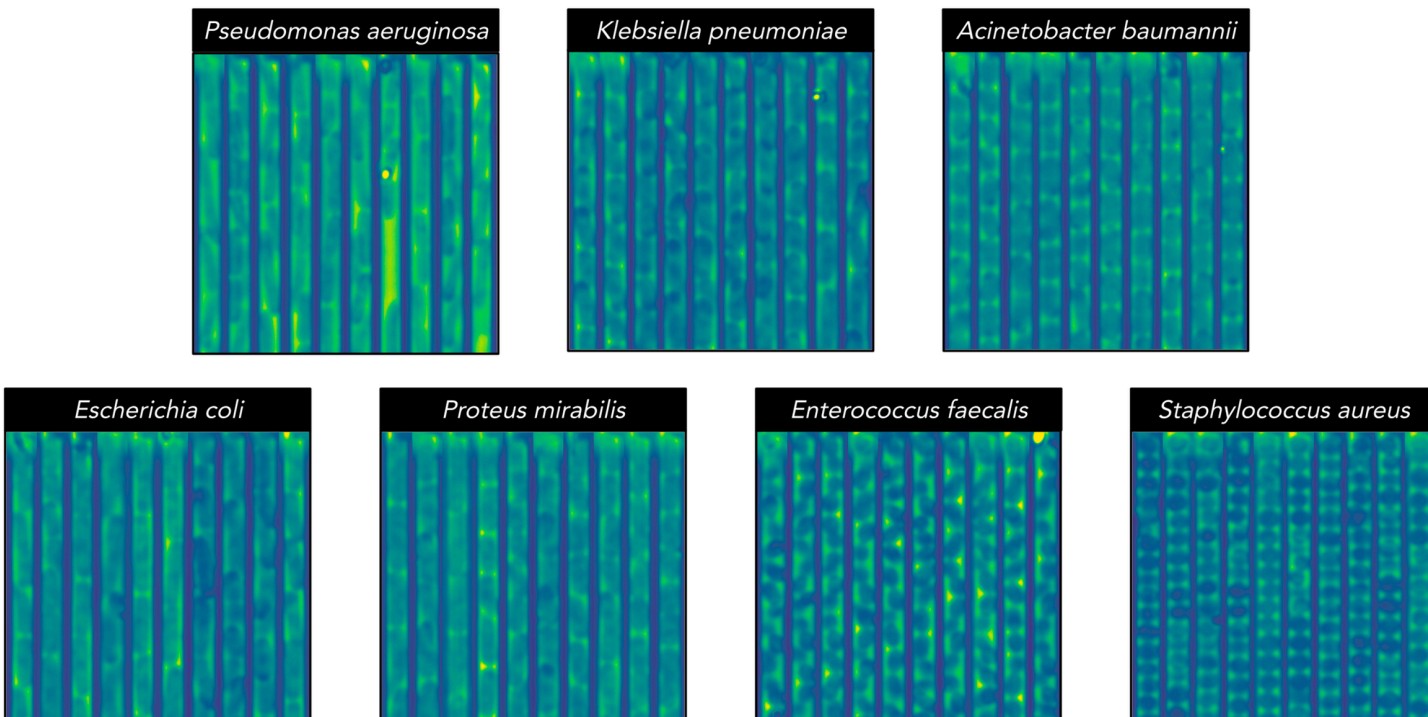

**Fig 13. Samples from the dataset.** Ten concatenated sample images (52x512 pixels) of each bacterial species taken from randomly selected traps. Visually distinguishing between the rod-shaped bacteria in this dataset is challenging, even for trained humans. Specifically, differentiating between *E. coli*, *P. mirabilis*, and *K. pneumoniae*. *P. aeruginosa* often has a characteristic thin-curved shape, while *A. baumannii* is comparatively shorter and thicker, however, these are not always reliable features. *S. aureus* had a larger diameter than *E. faecalis* for these isolates.

## 4.4 Image modifications

The networks were trained and evaluated using three different modifications on the phase-contrast data, visualized in Fig 14. The modifications are displayed side-by-side in S8 Video.

- **Default:** Original images without any modifications.
- **Mask:** Semantic segmentation masks pre-generated from the segmentation model (refer to the Segmentation Model section below).
- **Mean:** The pixel value of every pixel was replaced by the row-wise mean.

Furthermore, a final mode, "Segment" where the background was removed with a probability, was used as an augmentation step during training.

## 4.5 Stabilizing and cropping out traps

Our previous 4-species study [34] included software to crop and stabilize the phase contrast time-lapse, but it encountered frequent failures when applied to this dataset. Therefore, a new pipeline was developed using classical image processing methods to align, register, and crop out traps from the raw images. The main technique used in the pipeline was projecting the raw phase-contrast images on the vertical and horizontal axis. The resulting signal contained peaks where the traps (seen in Fig 1B and S9 Video) were located on the horizontal axis.

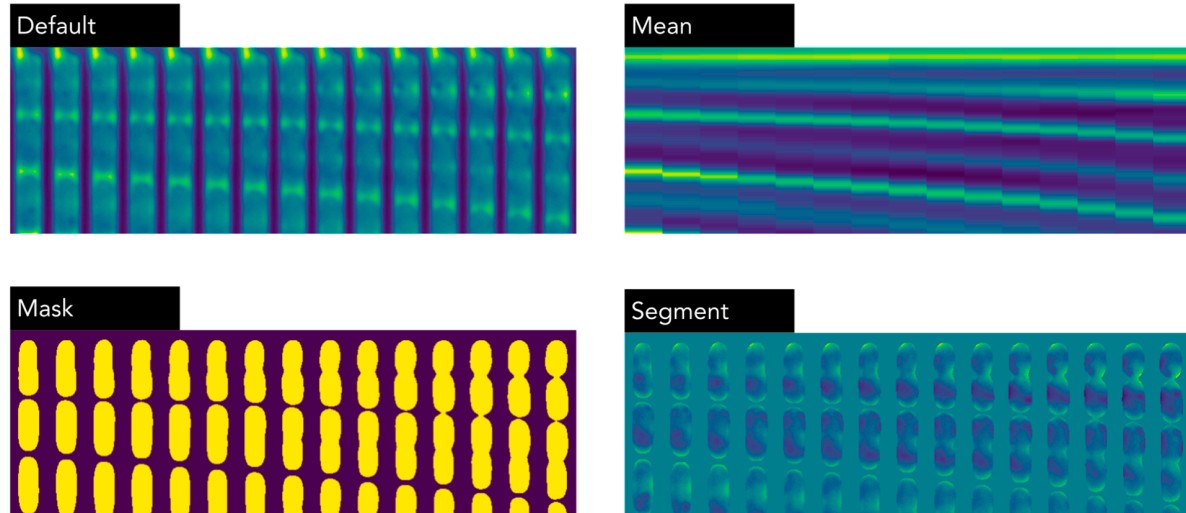

**Fig 14. Samples from the dataset.** Four modifications visualized for a part of a single trap (52x256 pixels, 15 frames). In the row-wise mean images, the cell height and division speed is visible, but not morphology and texture information. In the mask, no texture information is visible, and for the segment, all background is blocked out, aiming to reduce the risk of models overfitting to experimental settings.

The pipeline started with rotating the images since the chip had a slight tilt in the raw microscopy images, usually around 1-3 degrees. When the chip structure was perfectly horizontal, the peaks in the projections were at maximum values. The correct rotation was determined through an adaptive grid search, with progressively refined step sizes of 1, 0.1, and finally 0.05 degrees.

The rotation step was followed by a custom-developed rough registration method for aligning the phase-contrast time-lapse frames and fluorescence images. This was necessary because standard registration methods occasionally mixed up the vertical traps due to their regular striped pattern, especially when significant horizontal drift occurred during the time-lapse. The custom registration started by performing an adaptive threshold of the phase-contrast images revealing the chip structure. Each image in the phase-contrast stack was then shifted a number of pixels horizontally and vertically until aligned to the first image in the time-lapse (evaluating on the minimum of bitwise XOR between the binary images). This procedure was repeated, but instead registered each phase-contrast frame to the previous frame in the time-lapse. Then, the fluorescence images and phase-contrast images were projected on the horizontal axis, observing peaks that revealed the horizontal location of the traps. The fluorescence images were aligned to the last phase-contrast frame, shifting horizontally until the projections were aligned, evaluating using the absolute difference.

After this rotation and registration step, the phase-contrast images were projected onto the horizontal axis, and the horizontal location of each trap was extracted. The traps were cropped vertically, using a distance of 26 pixels from the center position on each side of a peak, extracting crops of 52 pixels width. A procedure was developed to crop the images horizontally right at the physical stop protruding from the side of a wall at the top of each microfluidic trap (visible as a circular blob in the top-right of each phase-contrast frame in Fig 2). The procedure utilized the horizontal asymmetry introduced in the images due to

this physical stop. The resulting crops had a size of 52x1500 pixels, showing a frame of a single trap. Finally, a fine registration was performed on each cropped-out phase-contrast stack using StackReg [55] using only translation (no scaling, rotating, or shear).

Barcodes were evenly laid out to locate traps in the chip, as seen on the sides and middle in Fig 1B and S9 Video. Typically, there were three barcodes visible in each position, but one on the side could go out of view if there was significant horizontal drift during capture. This drift introduced a problem since the barcodes generated peaks similar to a trap during horizontal projection. The cropped-out phase-contrast stacks extracted from the sides of the chip were projected onto the vertical axis to determine if they were barcodes. If the resulting 1D signal contained strong peaks with the width of a barcode dot, it was classified as a barcode. Also, the median pixel intensity was lower if the crop contained barcodes, which was used as an additional predictor.

The pipeline also generated debug images on each raw phase-contrast and fluorescence frame, highlighting the crop locations to ensure no errors occurred during the preprocessing.

## 4.6 Labeling tool

A program was developed to perform fluorescent signal aggregation and assign species labels to each trap. Examples of the four fluorescence channels are visualized on the very left in Fig 2. In the dataset, *E. coli* was visible in Cy3, *K. pneumoniae* in Cy3 and Cy5, *E. faecalis* in Alexa488, *P. aeruginosa* in Alexa488 and TxR (Texas Red), *A. baumannii* in Alexa488 and Cy3, *P. mirabilis* in Alexa488 and, *S. aureus* in Cy5 and TxR. Assigning a label proved challenging, as the images were often noisy and exhibited significant variations in fluorescence intensity across the fluorescent channels. The tool used standard background removal followed by applying a fluorescence segmentation model, selecting regions with bacterial illumination. The fluorescence images of the trap were then projected on the vertical axis, which converted the images into a 1D signal. At each vertical coordinate a species label was extracted using fine-tuned thresholds comparing the signal of each channel. The experiments were often performed with a few species per position, as shown in Table 1, and the labeling tool could use this prior information when assigning a label.

In total, 73706 traps were labeled using the labeling tool, and 38570 of these were retained containing single-species growth, discarding traps with multiple species or empty traps.

## 4.7 Data cleaning and label adjustments

A number of issues with the staining process were observed in the data. The cells in the final phase-contrast frame were not always vertically aligned with the fluorescence images. This misalignment may be due to cell shrinkage when dissolving the cell walls and the time lag between probe addition and fluorescence image capture.

Additionally, disparities in probe absorption were observed among the species; probes of one stain had a reduced absorption in certain species. This resulted in some fluorescent signals being notably dimmer or emitting no signal at all. Examples of these are shown in S15 Fig–S18 Figs.

After reviewing the output, 95 traps were discarded from the test set where the labeling tool clearly made an error, and the trap contained multiple species. Furthermore, 14 test traps were discarded due to either disconnecting from the top during growth or being empty for more than 15 consecutive frames (equivalent to 30 minutes). Additionally, 20 traps were relabeled as *K. pneumoniae* containing a dim but present signal in Cy5, and one trap was relabeled as *P. mirabilis*. Due to the low absorption of the fluorescent probe Alexa488, 25 samples were relabeled as *K. pneumoniae* that were previously unlabeled.

Similarly, a total of 355 traps were adjusted or discarded in the training set. Candidate mislabelings in both the training and test sets were identified by inspecting traps that the network consistently misclassified. These were then manually reviewed, and corrections were made only when clear labeling errors were confirmed. One intrinsic property of neural networks is their robustness to label noise [56]. Network misclassifications in both the training and test sets were often due to erroneous labeling. For transparency, all test-trap label adjustments and their rationale are displayed in the S1 Appendix.

## 4.8 Data quality score

After the labeling and discarding of traps being empty or containing multiple species, the training set was significantly unbalanced with 9719 *E. coli*, 7687 *P. aeruginosa*, 5957 *K. pneumoniae*, 4570 *A. baumannii*, 3408 *E. faecalis*, 2591 *P. mirabilis* and 2510 *S. aureus*. The ResNet 18 and Video ResNet R(1+2)D were then trained according to the following settings:

- Training on the full imbalanced dataset, with traps from each species randomly sampled for each batch in proportion to their fraction in the dataset, using a Random Weighted Sampler in Pytorch [57].
- Training on a balanced, capped dataset, containing a maximum of 2600 traps per species, selected randomly.
- Training on a balanced, capped dataset, containing a maximum 2600 of traps per species as previously, but first ranking the traps based on data quality outlined in Algorithm 1. For species with more than 2,600 samples (*E. coli*, *P. aeruginosa*, *K. pneumoniae*, *A. baumannii* and *E. faecalis*), this meant retaining the 2600 traps with the highest data quality scores.

The dataset was evaluated on the same hold-out test set as before, Experiment 17 in Table 1. The AUC results over thirty runs were for ResNet 18: $0.97 \pm 0.004$, $0.976 \pm 0.002$, and $0.987 \pm 0.001$, respectively, and for Video ResNet R(1+2)D: $0.983 \pm 0.003$, $0.984 \pm 0.004$ and $0.997 \pm 0.001$, visualized in Fig 11. Thus, using data quality scoring significantly improved performance. Consequently, all experiments conducted in this paper used a training set obtained by limiting at 2600 traps per species following the data scoring outlined in Algorithm 1.

## 4.9 Segmentation model

The segmentation model used in this paper was an ensemble model consisting of a U-Net [58] and a Segformer [59] model. A pixel was activated only when there was an agreement between both models at the pixel level (logical AND). The segmentation model was used to measure the vertical growth in each trap. Empirically, it was observed that U-Net performed better segmenting fine-grained local edges and textures, and the Segformer had better global awareness, presumably due to the global receptive field of the transformer architecture due to the self-attention mechanism [37].

Prior to using the segmentation model, classical image processing methods, such as histogram equalization [60] and Otsu's thresholding [61], were used to segment bacterial content in a trap. However, these methods proved unreliable, especially in the presence of brightness variations and out-of-focus areas across the image. Then Omnipose [62] was tested with the standard phase-contrast bacteria configuration but did not work, presumably due to the unusual elongated shape of the cropped trap images.

The data training for the ensemble model was obtained by first training a standard U-Net [58] model using segmentation masks published in [28], segmenting bacteria cells on 512x512

**Algorithm 1. Pseudocode for grading the data quality of each trap.** Traps with steady growth through many small steps, without abrupt jumps, and a clear fluorescent signal extending the full length of the trap in the final frame were assigned higher scores. Furthermore, the score was set to zero if a trap contained too many empty frames, the sum of the growth measurement array was negative, or the last frame had a growth measurement of less than 250 pixels.

**Require:** A An array of vertical growth measurements (in pixels) from a trap at each time-point (frame)

**Require:** F The vertical extent (in pixels) of fluorescent content after the final frame

<u>**Output:**</u> Score The data quality score of the trap

```
1:  Score ← 0
2:  Growth ← [ ]
3:  for i ← 2 to length(A) do
4:      current_growth ← A[i] – A[i – 1]
5:      Growth[i – 1] ← current_growth       ▷ Store the growth measurement
6:      Score ← Score + F × log(max(current_growth, 1))
7:  end for
8:
9:  n_empty ← 0
10: for i ← 1 to length(A) do  ▷ Count the number of empty frames
11:     if A[i] < 25 then
12:         n_empty ← n_empty + 1
13:     end if
14: end for
15: if ∑(Growth) < 0 or last(A) < 250 or n_empty > 25 then
16:     Score ← 0  ▷ Set the score to zero if abnormal events occur
17: end if
```

pixel crops. This model was then used to generate 50,000 binary segmentation masks from our phase-contrast dataset consisting of 52x1500 pixel crops. These masks were enhanced by using custom logic excluding regions with either too small area or minimal width that do not resemble bacterial cells. This dataset was then used to train the final ensemble model used in this paper.

The fluorescence images were often noisy, and standard background removal methods were frequently insufficient. Therefore, 1,000 segmentation masks of fluorescence images were manually labeled and used to train a segmentation model (the same Segformer U-Net ensemble outlined above) to identify regions with bacterial fluorescence. This fluorescence segmentation model was then integrated into the labeling tool.

## 4.10 Augmentations

The Albumentations library [48] was used with the augmentations ShiftScaleRotate, Random-BrightnessContrast, the erasings CoarseDropout, GridDropout and PixelDropout, Blur, HorizontalFlip, VerticalFlip, and RandomCrop. Furthermore, a custom-developed "Random video frame erasing" was developed that erased (set all pixels to zero) 1-2 consecutive frames at 1-3 positions in the time-lapse. Another custom augmentation method, "Segment", removed the background using pre-generated masks from the segmentation model. It was observed that RandomBrightnessContrast was of high importance, so it was always applied (100% probability). ShiftScaleRotate and horizontal and vertical flips were applied with a 50% probability,

while the remaining augmentations were applied with a 10% probability in the augmentation pipeline. Standard parameters were used for each augmentation with only minor adjustments.

## 4.11 Deep learning model training

The deep learning models were trained using the Pytorch [57] library, and the image models were obtained from the Pytorch Image Models (timm) library [63] and the video classification models from the Torchvision library [64]. The models were trained on images with a spatial size of 52x114 pixels, larger images caused the networks to overfit for this dataset. The image classification variants were chosen to have 11-12 million parameters: ResNet 18, Efficient-Net B3, RegNetY 016, and FastVit SA12. The video classification models ResNet R(2+1)D and Video ResNet R3D had around 30 million parameters.

The main idea behind ResNet variants is the use of residual connections, which mitigates performance degradation as networks become deeper [36]. Video ResNet R(2+1)D extends this concept to video processing. The model also decomposes standard 3D convolutions into separate 2D spatial convolutions followed by 1D temporal convolutions. In contrast, the Video ResNet R3D employs regular 3D convolutions to simultaneously detect spatial and temporal features.

RegNets [44] are a family of models proposed through an automated search, optimizing network width and depth to achieve optimal performance with a given number of floating point operations (FLOPs) budget for the inference of one sample. Similarly, EfficientNets [45] results from a comparable search to find optimal architectures. The EfficientNet models also use depth-wise separable convolutions to reduce the number of parameters and, thus, computational cost.

FastViT [46] is a recently developed vision transformer [37] incorporating several optimizations to enhance computational efficiency.

The models were trained in a standard supervised fashion using Adam optimizer [65], batch size 32, learning rate 0.0001, and cross-entropy loss function. A label smoothing of 0.15 was used to mitigate overfitting, distributing 15% of the probability mass to the other classes. This adjusted the one-hot encoded vectors to have a value of 0.85 for the true class and 0.025 for the false classes [49]. Weights were initialized by pretrained models by default. The image models were pretrained on the ImageNet dataset [66], and the video models were pretrained on the Kinetics-400 dataset [67].

During the image model training, a random non-empty frame was sampled from a trap. The trap was initially vertically cropped according to the vertical growth obtained from the segmentation model. This procedure ensured that the subsequent 52x114 pixel RandomCrop always contained bacterial content, avoiding empty sections of the trap. The video models were trained such that each sample was a 10-frame video clip randomly sampled from the time-lapse. The trap was similarly cropped vertically to ensure that at least 75% of the video frames were non-empty when extracting a video clip from the trap using RandomCrop.

The trap had an original width of 52 pixels and a height of 1500 pixels. During testing, a number of 52x114 pixel crops (image or video clips) were extracted from the traps in a tiling window arrangement, as seen in S1 and S2 Figs. The final output was obtained by summing the output logits from each crop. Initially, the output aggregation was performed by majority voting, but this did not improve performance. Furthermore, summing the logits simplified the Receiver Operating Characteristics (ROC) computations.

Time testing evaluations (Figs 3 and 7) were conducted by including all frames up to the target frame and then running testing. Video classification models can handle variable-length video inputs by applying global average pooling across the time dimension before the final

fully connected layer [68]. The image models were tested by including crops from all frames leading up to the target frame.

The video classification networks were trained for 150 epochs, and the image classification networks for 300 epochs. Nvidia A100 GPU:s were used for the training using MIG reservation for the less demanding downsampled models. Training times ranged from 1-2 hours for low-resolution ResNet 18 models to around 10 hours for the Video ResNet R(2+1)D at full resolution 52x114 pixels. Inference times were 0.00744 seconds per sample for Video ResNet R(2+1)D at full resolution and 6.351e-5 seconds per sample for ResNet 18 at full resolution, using a batch size of 32 on the A100 GPU.

The models were retrained five times for the scaling experiments, and 30 times for the augmentation ablation, and texture, morphology and division pattern experiments. Each retraining was conducted using a predetermined random seed for reproducibility.

## 4.12 Visualizations

All visualizations in this paper use Strip plot or Beeswarm plot from the Seaborn visualization library [69], which are scatter plots with jitter added to the points for readability. The trap visualizations use the Viridis colormap [70] to show the grayscale one-channel phase-contrast images.

## 4.13 Metrics

The precision and recall were calculated for each species i, using the confusion matrix (evaluated at argmax).

$$\text{Precision}_i = \frac{\text{TP}_i}{\text{TP}_i + \text{FP}_i} \quad i \in C \tag{1}$$

$$\text{Recall}_i = \frac{\text{TP}_i}{\text{TP}_i + \text{FN}_i} \quad i \in C \tag{2}$$

Where $TP_i$ and $FP_i$, and $FN_i$ are the number of true positives, false positives and false negatives for the respective species i, C are the set of classes.

Then, Receiver Operating Characteristic (ROC) curves and the corresponding Area Under the Curve (AUC) were computed using varying thresholds. AUC was selected as the evaluation metric for all experiments in this paper because it provides a comprehensive and stable evaluation across varying thresholds. Using a single metric for all tests simplifies comparisons. We are primarily interested in the relative changes in AUC across different modifications and experiments, rather than absolute performance values, to understand what works. The AUC was obtained using the Scikit-Learn library [71] and calculated for each species in a one-vs-rest fashion, treating it as a binary classification problem. The average AUC reported used the micro-averaged ROC curve obtained using the following formula, evaluated at each threshold:

$$\text{TPR} = \frac{\sum_i \text{TP}_i}{\sum_i (\text{TP}_i + \text{FN}_i)}, \quad i \in C \tag{3}$$

$$\text{FPR} = \frac{\sum_i \text{FP}_i}{\sum_i (\text{FP}_i + \text{TN}_i)} \quad i \in C \tag{4}$$

### 4.14 Spatial downsampling

The models were trained and tested using downsampled images using the Lánczos method [47]. In total, 28 downsampling steps were used: 52x114, 48x105, 44x96, 40x88, 36x79, 32x70, 28x61, 26x57, 24x53, 22x48, 20x44, 19x42, 18x39, 17x37, 16x35, 15x33, 14x31, 13x28, 12x26, 11x24, 10x22, 9x20, 8x18, 7x15, 6x13, 5x11, 4x9 and 3x7 pixels.

### 4.15 Use of artificial intelligence tools and technologies

The tools Grammarly and ChatGPT were used for grammar checking, as a synonym book, and as rephrasing tools when writing this article. No original content was generated by the models. Furthermore, the coding assistant GitHub Copilot was used to generate type annotations and docstrings for the software in the released replication package.

## Supporting information

**S1 Appendix. Testset adjustments and data cleaning** The document shows all traps that either had the label manually changed or were removed from the test set after processing of the labeling tool.
(PDF)

### Crop extraction

The following figures show crop extraction from a time-lapse. **S1 Fig. Crops extracted from a trap for image classification.** The vertical growth extent of the bacteria in each trap was measured using the segmentation model, indicated by the horizontal markings on the tip of the lowest cell in each trap. Crops were extracted in a tiling-window fashion, shown by outlined yellow squares. In total, 202 52x114 pixel crops were extracted from this trap. The criteria for including a crop is that at least 25% of its vertical extent contains bacteria.
(PNG)

**S2 Fig. Crops extracted from a trap for video classification.** The vertical growth extent of the bacteria in each trap was measured using the segmentation model, indicated by the horizontal markings on the tip of the lowest cell in each trap. Each video crop is outlined with a yellow square. The vertical dotted lines between adjacent frames indicate they are part of the same video crop. In total, five 52x114x35 pixel video crops were extracted from this trap. The criteria for including a video crop is that at least 75% of the frames contain bacterial content.
(PNG)

### Misclassifications

**S3 Fig. Trap misclassified by Video ResNet R(2+1)D at full resolution.** *E. coli* and *P. mirabilis* are visually very similar in shape, both being rods. There is some overcrowding in the trap with overlapping cells.
(PNG)

**S4 Fig. Trap misclassified by Video ResNet R(2+1)D at full resolution.** *E. coli* and *K. pneumoniae* are visually very similar in shape, both being rods. There is some overcrowding in the trap with overlapping cells at the end of the time-lapse.
(PNG)

**S5 Fig. Trap misclassified by Video ResNet R(2+1)D at full resolution.** K. penumaniae and *E. coli* are visually very similar in shape, both being rods. There is some dislocation in the trap at the end of the time-lapse, causing one of the video-crops to have empty frames. (PNG)

**S6 Fig. Trap misclassified by Video ResNet R(2+1)D at full resolution.** *S. aureus* and *E. faecalis* are visually very similar in shape, both being cocci. There is a dislocation in the top, one of the video crops has only empty frames. This trap should have been discarded according to the discarding criteria (no dislocation from the top of the trap), but it was missed during the test set inspection. (PNG)

**S7 Fig. Trap misclassified by Video ResNet R(2+1)D at full resolution.** The model mistakes the circular physical stop at the top of the trap for *E. faecalis*. This trap should have been discarded according to the discarding criteria since the trap is empty up to frame 15 (cells need to be loaded within 30 minutes), but it was missed during the test set inspection. (PNG)

**S8 Fig. Trap misclassified by Video ResNet R(2+1)D at full resolution.** A rod cell is present at the beginning of the time-lapse, which then dissolves and is not visible in the fluorescence images. (PNG)

**S9 Fig. Trap misclassified by Video ResNet R(2+1)D at downsampled 5x11 resolution.** *P. mirabilis* and *E. coli* are visually very similar in shape, both being rods. Classifying at very low resolution is significantly more challenging. (PNG)

**S10 Fig. Trap misclassified by Video ResNet R(2+1)D at downsampled 5x11 resolution.** *K. pneumoniae* and *E. coli* are visually very similar in shape, both being rods. Classifying at very low resolution is significantly more challenging. (PNG)

**S11 Fig. Trap misclassified by Video ResNet R(2+1)D at downsampled 5x11 resolution.** *A. baumannii* and *E. coli* are visually very similar in shape, both being rods. Classifying at very low resolution is significantly more challenging. (PNG)

**S12 Fig. Trap misclassified by Video ResNet R(2+1)D at downsampled 5x11 resolution.** This trap was also misclassified by Video ResNet in the full resolution. *E. coli* and *P. aeruginosa* are visually very similar in shape, both being rods. Classifying at very low resolution is significantly more challenging. (PNG)

**S13 Fig. Trap misclassified by Video ResNet R(2+1)D at downsampled 5x11 resolution.** This trap was also misclassified by Video ResNet in the full resolution. *S. aureus* and *E. faecalis* are visually very similar in shape, both being cocci. There is a dislocation in the top, one of the video crops has only empty frames. This trap should have been discarded according to the discarding criteria (no dislocation from the top of the trap), but it was missed during the test set inspection. Classifying at very low resolution is significantly more challenging. (PNG)

**S14 Fig. Trap misclassified by Video ResNet R(2+1)D at downsampled 5x11 resolution.** *P. aeruginosa* and *A. baumannii* are visually very similar in shape, both being rods. This trap should have been discarded according the discarding criteria as it is empty until frame 25

(cells must be loaded within 30 minutes), but it was missed during the test set inspection. Classifying at very low resolution is significantly more challenging.
(PNG)

## Combination staining issues

The following figures show examples of challenges in assigning labels to traps due to irregularities in the staining process.

**S15 Fig. K. pneumonie.** For K. pneumonie the cells are visible in Cy3 and Cy5 in the fluorescent staining.
(PNG)

**S16 Fig. K. pneumonie.** The cells should be visible in Cy3 and Cy5, but there was almost no absorption of Cy5 fluorophores. The trap originated from the same experiment and chip section as S15 Fig, which contained only *K. pneumoniae*, allowing the labeling tool to infer the correct label.
(PNG)

**S17 Fig. *P. aeruginosa*.** The species should be visible in Alexa488 and TxR, but there was no absorption of Alexa488 fluorophores. It can not be *S. aureus* (Cy5, TxR) being a cocci; hence, the label was manually assigned.
(PNG)

**S18 Fig. *S. aureus*.** The species should be visible in Cy5 and TxR, but there was no absorption of Cy5 fluorophores. The chip section contained only *S. aureus*, allowing the labeling tool to infer the correct label.
(PNG)

## Videos

The following video clips show the same trap with all the downsampling steps used in the resolution scaling experiments. The number in the top row indicates the pixel width.

**S1 Video. *P. aeruginosa*.**
(MP4)

**S2 Video. *E. coli*.**
(MP4)

**S3 Video. *K. pneumoniae*.**
(MP4)

**S4 Video. *A. baumannii*.**
(MP4)

**S5 Video. *E. faecalis*.**
(MP4)

**S6 Video. *P. mirabilis*.**
(MP4)

**S7 Video. *S. aureus*.**
(MP4)

The following are other supporting videos for the article.

**S8 Video. Image distortion modes.** The two image distortion modes "Mean" and "Mask" are visualized for a single trap on the left, and the default phase-contrast images on the right.
(MP4)

**S9 Video. Microfluidic mother machine.** Raw unstabilized phase-contrast microscopy time-lapse of a chip position in the microfluidic mother machine followed by fluorescence microscopy capturing four stains, Alexa488, Cy3, Cy5, and Texas Red (TxR). The single-channel raw grayscale images are shown here (no color map applied).
(MP4)

## Author contributions

**Conceptualization:** Erik Hallström.

**Data curation:** Vinodh Kandavalli.

**Formal analysis:** Erik Hallström.

**Funding acquisition:** Carolina Wählby.

**Investigation:** Erik Hallström.

**Methodology:** Erik Hallström.

**Software:** Erik Hallström.

**Supervision:** Anders Hast.

**Validation:** Erik Hallström.

**Visualization:** Erik Hallström.

**Writing – original draft:** Erik Hallström.

**Writing – review & editing:** Erik Hallström, Carolina Wählby, Anders Hast.

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
