## [Decision Letter · Decision Letter 0]

16 Jun 2025

PONE-D-25-18176Rapid label-free identification of seven bacterial species using microfluidics, single-cell time-lapse phase-contrast microscopy, and deep learning-based image and video classificationPLOS ONE

Dear Dr. Hallström,

Thank you for submitting your manuscript to PLOS ONE. After careful consideration, we feel that it has merit but does not fully meet PLOS ONE’s publication criteria as it currently stands. Therefore, we invite you to submit a revised version of the manuscript that addresses the points raised during the review process.

We look forward to receiving your revised manuscript.

Kind regards,

Baeckkyoung Sung, Ph.D.

Academic Editor

PLOS ONE

Journal Requirements:

4. Please note that your Data Availability Statement is currently missing the repository name. If your manuscript is accepted for publication, you will be asked to provide these details on a very short timeline. We therefore suggest that you provide this information now, though we will not hold up the peer review process if you are unable.

Additional Editor Comments:

The authors are suggested to revise the manuscript in line with the reviewers' comments.

Reviewers' comments:

Reviewer's Responses to Questions

**Comments to the Author**

1. Is the manuscript technically sound, and do the data support the conclusions?

Reviewer #1: Yes

Reviewer #2: Yes

Reviewer #3: Yes

2. Has the statistical analysis been performed appropriately and rigorously? 

Reviewer #1: No

Reviewer #2: Yes

Reviewer #3: Yes

3. Have the authors made all data underlying the findings in their manuscript fully available?

Reviewer #1: Yes

Reviewer #2: Yes

Reviewer #3: No

4. Is the manuscript presented in an intelligible fashion and written in standard English?

Reviewer #1: Yes

Reviewer #2: Yes

Reviewer #3: Yes

5. Review Comments to the Author

Reviewer #1: The manuscript presents a novel approach for bacterial species identification using microfluidics, time-lapse microscopy and deep learning. While the methodology is technically sound, several critical issues must be addressed before publication can be considered.

The authors emphasize their "label-free rapid identification" as a key innovation. However, the previous work already demonstrated this approach for 4 species. Simply extending to 7 species without significant methodological advancement is insufficient. No clinical isolates were tested, only lab strains. The claimed clinical applicability remains unproven. The "one-hour identification" claim lacks proper context. No comparison with standard methods, such as MALDI-TOF or qPCR in terms of total processing time. No cost analysis provided despite emphasizing "low-resource settings". The microfluidic system requires high bacterial concentrations. No solution is provided for processing complex clinical samples.

Only single strains per species were tested. No evaluation of medium effects on optical phenotypes. Gram-positive rods and Gram-negative cocci completely omitted.

It is recommended to clearly differentiating from previous work, demonstrating clinical isolate testing, and adding comparison tables.

Reviewer #2: This study presents a rapid diagnostic approach for identifying bacterial pathogens within one hour using phase-contrast time-lapse imaging of single bacterial cells trapped in a microfluidic "mother machine". By training deep learning models —specifically Convolutional Neural Networks (CNNs) and Vision Transformers (ViTs) — on these videos, the authors showed that they can accurately classify seven common infection-causing species.

The method achieves high performance and leverages spatiotemporal features such as cell texture and morphology. It also supports real-time evaluation as new frames are captured and can be integrated with phenotypic antibiotic susceptibility testing (AST). Despite its promise, practical challenges like isolating bacteria directly from blood and validating against clinical strains remain. This proof-of-concept marks a step toward real-time diagnostics for acute infections.

I consider this study being publishable in PLOS ONE, as this jounrnal publishes solid, reproducible science — regardless of novelty or expected impact. I am saying this because this study is a follow-up study of an earlier study by these authors published in PLOS Computational Biology (Reference 21). What makes this study different from their previous work is that they (i) study real-time performance by capturing additional frames during testing, (ii) investigate the role of training set size, (iii) data quality, and (iv) data augmentation, as well as (v) the contribution of texture and morphology to performance.

The manuscript is written fairly technically and recent biological advances are missing from the manuscript. I ask the authors to include and discuss recent advances on antibiotic susceptibility testing into their manuscript. In fact, looking at pubmed (https://pubmed.ncbi.nlm.nih.gov/?term=antibiotic+susceptibility+testing&sort=pubdate), work in this area is exponentially increasing during the last years, while the authors do not cite any of those studies after the year 2022. This should be improved by discussing their own approach in the context of other approaches (for example, see this recent work: https://doi.org/10.1016/j.snb.2024.136866 and the references discussed therein).

I recommend a minor revision for improving the missing references to recent work being implemented in the Introduction or Discussion section.

Reviewer #3: Hallström et al. present a thorough work on bacterial species classification using Deep learning in a Mother Machine. I find it especially interesting that they investigate the possibility of just using masks do do the classification with high accuracy. This suggests that the method should work well under different conditions as long as good segmentation is possible. Maybe this should be emphasized a bit more in the Discussion, but i leave it up to the authors if they want to do that.

My main issue is that it is not clearly stated how and where the Code and Data can be obtained without restrictions. In the Data Availability section the authors have just stated that "Yes - all data are fully available without restriction". It's even mentioned by PLOS that "‘data available on request from the author’ is not sufficient". I saw that the dataset is from a different publication where the data availability was "data is available upon reasonable request from the author’, but as there is at least one co-author on both publications so please make an effort to make both Code and Data available. I would personally suggest https://bioimage.io/#/, but if the authors want to use other repositories that is also fine.

For the species detection based purely on segmentation masks, it would be interesting to see how well Linear Discriminant Analysis would perform just based on standard shape characteristics that you would get from ImageJs "Analyze Particles" or similar methods.

6. PLOS authors have the option to publish the peer review history of their article (what does this mean?). If published, this will include your full peer review and any attached files.

Reviewer #1: No

Reviewer #2: No

Reviewer #3: **Yes: **Carl-Magnus Svensson

---

## [Author Response · Author response to Decision Letter 1]

30 Jun 2025

The responses to the reviewers are provided in the attached file "Response_to_the_reviewers_letter.pdf"

To the Editor,

1. We have reviewed the file names, layout etc and believe that we are compliant with the journal’s requirements.

2,4:

The data repository is included in the Data Availability Statement, and a data citation is provided in the manuscript:

"""

The authors confirm that all data underlying the findings are fully available without restriction. A replication package is available at https://doi.org/10.5281/zenodo.13321089 containing all image data and software to reproduce the experiments, generate output metrics and build the graphs in the article.

"""

3. None of the co-authors have any economic interest in the results. The funding information and grant number are correct.

Best regards,

Erik

---

## [Decision Letter · Decision Letter 1]

30 Jul 2025

Rapid label-free identification of seven bacterial species using microfluidics, single-cell time-lapse phase-contrast microscopy, and deep learning-based image and video classification

PONE-D-25-18176R1

Dear Dr. Hallström,

We’re pleased to inform you that your manuscript has been judged scientifically suitable for publication and will be formally accepted for publication once it meets all outstanding technical requirements.

Kind regards,

Baeckkyoung Sung, Ph.D.

Academic Editor

PLOS ONE

Additional Editor Comments (optional):

The manuscript has well been revised in accordance to the reviewers' comments.

Reviewers' comments:

Reviewer's Responses to Questions

**Comments to the Author**

1. If the authors have adequately addressed your comments raised in a previous round of review and you feel that this manuscript is now acceptable for publication, you may indicate that here to bypass the “Comments to the Author” section, enter your conflict of interest statement in the “Confidential to Editor” section, and submit your "Accept" recommendation.

Reviewer #1: All comments have been addressed

Reviewer #2: All comments have been addressed

2. Is the manuscript technically sound, and do the data support the conclusions?

Reviewer #1: Yes

Reviewer #2: Yes

3. Has the statistical analysis been performed appropriately and rigorously? 

Reviewer #1: Yes

Reviewer #2: Yes

4. Have the authors made all data underlying the findings in their manuscript fully available?

Reviewer #1: Yes

Reviewer #2: Yes

5. Is the manuscript presented in an intelligible fashion and written in standard English?

Reviewer #1: Yes

Reviewer #2: Yes

6. Review Comments to the Author

Reviewer #1: I think all my concerns have been addressed and the manuscript is recommended to be accepted by PLOS ONE.

Reviewer #2: The authors have done a great job in addressing all issues raised in my review. The manuscript can now be accepted for publication.

7. PLOS authors have the option to publish the peer review history of their article (what does this mean?). If published, this will include your full peer review and any attached files.

Reviewer #1: No

Reviewer #2: No

---

## [Editor Report · Acceptance letter]

PONE-D-25-18176R1

PLOS ONE

Dear Dr. Hallström,

I'm pleased to inform you that your manuscript has been deemed suitable for publication in PLOS ONE. Congratulations! Your manuscript is now being handed over to our production team.

Kind regards,

on behalf of

Dr. Baeckkyoung Sung

Academic Editor

PLOS ONE